# Effective MoE-based LLM Compression by Exploiting Heterogeneous Inter-Group Experts Routing Frequency and Information Density

Zhendong Mi [1]   Yixiao Chen [2]   Pu Zhao [2]   Xiaodong Yu [1]   Hao Wang [1]   Yanzhi Wang [2]   Shaoyi Huang [1]

## Abstract

Mixture-of-Experts (MoE) based Large Language Models (LLMs) have achieved superior performance, yet the massive memory overhead caused by storing multiple expert networks severely hinders their practical deployment. Singular Value Decomposition (SVD)-based compression has emerged as a promising post-training technique; however, most existing methods apply uniform rank allocation or rely solely on static weight properties. This overlooks the substantial heterogeneity in expert utilization observed in MoE models, where frequent routing patterns and intrinsic information density vary significantly across experts. In this work, we propose RFID-MoE, an effective framework for MoE compression by exploiting heterogeneous Routing Frequency and Information Density. We first introduce a fused metric that combines expert activation frequency with effective rank to measure expert importance, adaptively allocating higher ranks to critical expert groups under a fixed budget. Moreover, instead of discarding compression residuals, we reconstruct them via a parameter-efficient sparse projection mechanism to recover lost information with minimal parameter overhead. Extensive experiments on representative MoE LLMs (e.g., Qwen3, DeepSeekMoE) across multiple compression ratios demonstrate that RFID-MoE consistently outperforms state-of-the-art methods like MoBE and $D^2$-MoE. Notably, RFID-MoE achieves a perplexity of 16.92 on PTB with the Qwen3-30B model at a 60% compression ratio, reducing perplexity by over 8.0 compared to baselines, and improves zero-shot accuracy on HellaSwag by approximately 8%.

[1] Department of Computer Science, Stevens Institute of Technology, Hoboken, New Jersey, USA   [2] Northeastern University, Boston, Massachusetts, USA . Correspondence to: Shaoyi Huang <shuang59@stevens.edu>.

*Proceedings of the 43$^{rd}$ International Conference on Machine Learning*, Seoul, South Korea. PMLR 306, 2026. Copyright 2026 by the author(s).

## 1. Introduction

The landscape of Natural Language Processing (NLP) has been reshaped by Transformer-based LLMs with strong performance across tasks (Vaswani et al., 2017; Zhan et al., 2024b;a; Shen et al., 2025b; Kong et al., 2025b; Shen et al., 2025c; Zhao et al., 2024). To balance massive capacity demands with computational constraints, recent architectures adopt Mixture-of-Experts (MoE) designs using specialized expert networks with sparse activation (Team et al., 2025), enabling models like DeepSeek (Liu et al., 2024a) and Qwen3 (Yang et al., 2025a) to scale effectively (Hoffmann et al., 2022). However, MoE models face severe deployment challenges from substantial memory overhead (Tang et al., 2024; Kong et al., 2025a; Zhong et al., 2024; Hwang et al., 2024). For instance, Qwen3-235B-A22B-2507 exceeds the 320GB capacity of an 8×40GB A100 cluster, causing out-of-memory failures (Song et al., 2024).

Existing MoE compression can be categorized into three categories: expert pruning, expert merging, and decomposition. Pruning removes experts but often loss specialized knowledge, degrading performance on domain-specific tasks (Xie et al., 2024; Lu et al., 2024; Gu et al., 2025). Expert merging consolidates experts assuming functional redundancy, but experts often exhibit distinct specializations (Li et al., 2023; Chen et al., 2024; Liu et al., 2024b). Decomposition methods, particularly SVD-based approaches (Li et al., 2025c;b; Chen et al., 2025), compress expert weights without reducing expert count, better preserving capacity. However, existing SVD-based methods face several limitations.

**Challenge 1: Employing uniform rank without considering imbalanced expert utilization may lead to suboptimal compression performance.** Expert utilization in MoE models exhibits significant non-uniformity across practical inference workloads, with some experts frequently activated while others remain largely underutilized. Despite this heterogeneity, existing compression methods predominantly assign uniform ranks to all experts or expert groups (Chen et al., 2025; Li et al.; He et al., 2024), leading to suboptimal performance. Even approaches that consider differences between experts typically rely exclusively on static weight properties (Leyang et al., 2025; Li et al., 2025b), neglecting dynamic activation patterns during inference. While

incorporating routing frequency as a criterion appears intuitive, it presents a fundamental challenge: activation distributions in large MoEs are often severely skewed, with some experts never activated, therefore with 0 rank allocated. Thus, compression that allocate ranks based purely on routing frequency degenerate into expert pruning, risking severe performance degradation on specialized tasks where infrequently-activated experts encode critical domain-specific knowledge (Chen et al., 2025; Gu et al., 2025).

**Challenge 2: The compression residual is non-negligible.**
In SVD-based compression, the fundamental objective is to minimize reconstruction error between the compressed and original matrices, thereby preserving model performance. Standard SVD-based methods retain the dominant singular components while directly discarding the residual term (Li et al., 2024; Chen et al., 2025), implicitly assuming negligibility of the approximation error. However, this assumption deserves closer examination. In particular, three important questions remain underexplored: (i) whether these residuals are genuinely negligible in magnitude and impact; (ii) what statistical properties characterize their distribution, and (iii) whether they can be reconstructed in a parameter-efficient manner instead of being discard entirely.

To address the challenges, we propose **RFID-MoE**, an effective SVD-based expert compression method tailored for Mixture-of-Experts (MoE) models that exploits the heterogeneous **R**outing **F**requency and **I**nformation **D**ensity across expert groups. Specifically, we develop adaptive rank allocation strategy that joint considers expert routing frequency and information density. By allocating larger ranks to frequently routed and information-rich experts while assigning fewer ranks to less critical ones, our method better preserves task-relevant performance under a fixed compression budget. This approach avoids the suboptimal uniform rank assignment or purely static weight-based rank allocation used in prior decomposition-based methods. Additionally, rather than discarding the compression residual, we reconstruct it using a parameter-efficient mechanism. This design enables RFID-MoE to retain complementary information that cannot be captured by low-rank approximation alone, further enhancing performance with only marginal additional parameters. Our contributions are summarized as follows:

- We propose RFID-MoE, an effective rank allocation compression method for MoEs that adaptively distributes compression rank budgets across experts to better preserve information with superior performance.

- We propose using the effective rank to analyze the information density of each expert group. Building on this, we develop a novel rank allocation metric that combines effective rank with normalized routing frequency to better preserve information for critical expert groups during compression.

- We analyze residual distribution patterns and propose a parameter-efficient reconstruction method to recover information lost in low-rank approximation.

We conduct extensive experiments on representative MoE-based LLMs including DeepSeekMoE-16B-Base, Qwen3-30B-A3B-2507, and Ling-mini-2.0. RFID-MoE achieves state-of-the-art performance across diverse tasks and compression ratios, e.g., surpassing the best baseline by 8.0 PPL on PTB and 8% on HellaSwag at 60% compression ratio.

## 2. Preliminaries and Motivation

### 2.1. Mixture-of-Experts Architecture

In Transformer-based models, the standard dense Feed-Forward Network (FFN) can be replaced with a MoE layer comprising a routing module and $n$ independent expert networks. Following recent open-source models (Shazeer, 2020), individual experts are typically implemented as FFNs with SwiGLU activation.

For an input token representation $x \in \mathbb{R}^d$, the computation performed by the $i$-th expert $E^i(x)$ consists of three linear transformations: an up-projection, a gate-projection, and a down-projection. The forward pass is formulated as:

$$E^i(x) = W_{down}^i \cdot \left( W_{up}^i x \odot \text{SiLU}(W_{gate}^i x) \right) \quad (1)$$

where $\odot$ denotes the element-wise product. $W_{up}^i, W_{gate}^i \in \mathbb{R}^{p \times d}$ and $W_{down}^i \in \mathbb{R}^{d \times p}$ are the three learnable weight matrices, where $p$ is the intermediate dimension.

The router module $G$ employs a learnable weight matrix $W_g \in \mathbb{R}^{n \times d}$ to compute routing probabilities and select the top-$K$ experts with the highest affinity scores:

$$G(x) = \text{TopK}(\text{Softmax}(W_g x)) \quad (2)$$

The final output of the MoE layer is computed by aggregating the outputs of the selected experts, weighted by their corresponding routing probabilities. This computation is performed independently for each token:

$$y = \sum_{i=1}^{K} G^i(x) E^i(x) \quad (3)$$

where $K$ denotes the number of activated experts per token.

### 2.2. SVD-based compression with basis sharing for MoE

Weight matrices often exhibit significant redundancy (Wang et al., 2024; Shen et al., 2025b;e; 2024; Chen et al., 2025; Mi et al., 2025), enabling joint SVD decomposition across multiple matrices for shared basis representations. Following (Wang et al., 2024; Chen et al., 2025), weight matrix

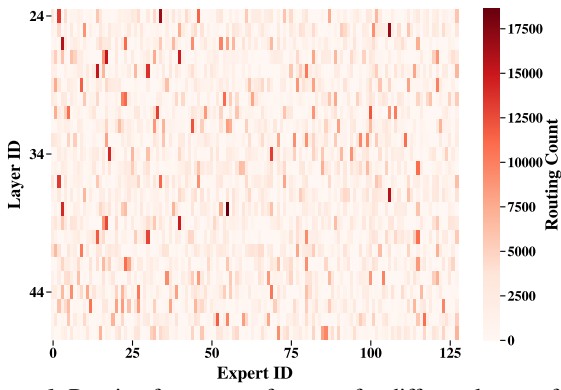

*Figure 1.* Routing frequency of experts for different layers of Qwen3-30B-A3B-2507 from layer 24 to 48.

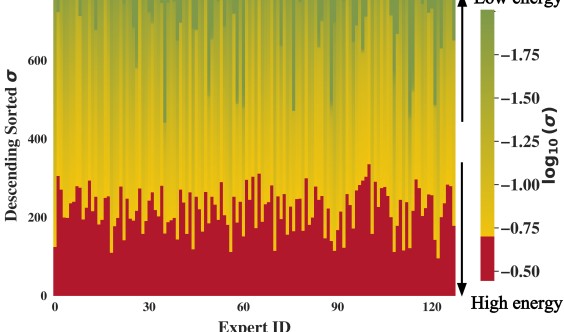

*Figure 2.* Residual singular value spectrum of gate matrix among experts at layer 0 in Qwen3-30B-A3B-2507. Each column corresponds to one expert, and each row represents the singular-value rank value of $\Delta W$.

$\mathbf{W}_i \in \mathbb{R}^{p \times d}$ of the $i$-th expert in group $j$ can be decoupled into an expert-specific projection $\mathbf{A}_i$ and a shared component $\mathbf{B}_j$. Rather than using $\mathbf{B}_j$ independently, we synthesize them from a global dictionary of basis matrices $\mathbf{B}_j \in \{\mathbf{B}_1, \ldots, \mathbf{B}_m\}$ shared across the layer, where $m \ll n$ ($m$ and $n$ denote the number of groups and experts, respectively). The expert weight can be approximated as:

$$\hat{\mathbf{W}}_i \approx \mathbf{A}_i \cdot \phi \left( \sum_{j=1}^{m} \alpha_{i,j} \mathbf{B}_j \right) \qquad (4)$$

where $\alpha_{i,j}$ is the learnable mixing coefficient for the $i$-th expert and the $j$-th basis, and $\phi(\cdot)$ is an activation function (e.g., SiLU) that enhances low-rank expressiveness.

Inspired by information distribution in LLMs (Meng et al., 2023), Up and Gate projection matrices are compressed and trained, while the Down projection matrices are remain dense to preserve semantic information. The training procedure reduces to a simple matrix reconstruction problem and can be performed without requiring training dataset. We freeze the original model parameters and optimize the factorized components ($\mathbf{A}_i$, $\mathbf{B}_j$, and coefficients $\alpha_{i,j}$) by minimizing the Frobenius loss norm $\mathcal{L}$ between original weights $\mathbf{W}_i$ and approximations $\hat{\mathbf{W}}_i$:

$$\mathcal{L} = \sum_{i=1}^{n} \left\| \mathbf{W}_i - \mathbf{A}_i \cdot \phi \left( \sum_{j=1}^{m} \alpha_{i,j} \mathbf{B}_j \right) \right\|_F^2 \qquad (5)$$

We can optimize using standard gradient-based methods (e.g., Adam), which offer superior stability over alternating optimization techniques for this non-linear formulation.

### 2.3. Motivation

**Imbalanced Expert Utilization.** In MoE models, experts exhibit non-uniform activation patterns during inference. Under a fixed routing mechanism, different experts receive substantially different numbers of tokens, creating highly imbalanced routing patterns. To further visualize the issue,

we sample 512 sequences from the WikiText-2 dataset with a sequence length of 1,024 tokens, and record expert routing frequency across all tokens at each layer. As shown in Figure 1, the expert routing frequencies are extremely imbalanced. While a small subset of experts is used very frequently, many experts are rarely selected. For example, in layer 38, expert 55 is routed 20388 times, while expert 102 is routed 0 times, indicating a pronounced imbalance in expert utilization. Similar patterns consistently emerge across different layers, suggesting that treating all experts equally during grouping and SVD compression is suboptimal.

**The Compression Residual is Non-negligible.** Most prior studies (Chen et al., 2025; Li et al., 2025c) truncate singular values by magnitude, retaining only the high-energy components and removing the residual term $\mathbf{R}$. However, the energy contained in the residual is often non-negligible. Figure 2 shows the residual energy distribution for gate matrices from experts in the first layer of Qwen3-30B-A3B-2507 at a compression ratio of 40%. A substantial portion of the residual singular values exceeds $0.1\times$ the maximum singular value (approximately 2.1 for layer-0 gate matrices), indicating that the elimination of these residual components causes significant information loss. Moreover, we observe that the singular value spectrum exhibits a characteristic decay pattern: values decrease rapidly from the middle of the spectrum, falling below 0.01 and eventually below 0.001 toward the tail. This pattern indicates that while the residual $\mathbf{R}$ contains non-negligible energy, its information concentrates in a low-dimensional subspace, enabling efficient reconstruction with minimal additional parameters.

## 3. Method

This section presents RFID-MoE, an effective compression method for MoE-based LLMs. First, we propose a routing frequency metric (Section 3.1) to partition experts into groups with similar activation frequency, enabling effective basis sharing within each group. Second, we introduce an effective rank metric (Section 3.2) to quantify the informa-

tion density of each expert group. We then integrate these two metrics to guide adaptive rank allocation across expert groups (Section 3.3). Finally, we propose a parameter-efficient residual reconstruction method to recover information lost during low-rank approximation (Section 3.4).

### 3.1. Routing Frequency of Expert Groups

Specifically, we first select a calibration dataset of moderate size to collect activation statistics, and compute the activation frequency $z_i$ for each expert $\mathbf{W}_i$ at every layer. We define the activation frequency of expert $\mathbf{W}_i$ as $F_i$ given by

$$F_i = \frac{z_i}{\sum_{b=1}^{n} z_b} \tag{6}$$

Given the activation frequencies $\{F_i\}_{i=1}^{n}$ of the $n$ experts, we first sort all experts in descending order of their activation frequencies. Based on this ordering, the experts are grouped sequentially, and SVD-based shared basis decomposition is performed within each group of expert matrices $\mathbf{W}_g \in \mathbb{R}^{kp \times d}$ ($k = n/m$). Subsequently, for each group-wise concatenated expert matrix $\mathbf{W}_g$, we compute the corresponding group activation frequency $F_g$:

$$F_g = \frac{\sum_{l=1}^{k} z_l}{\sum_{b=1}^{n} z_b} \tag{7}$$

### 3.2. Effective Rank for Expert Group Information Density Evaluation

A natural strategy for evaluating expert importance is based on routing frequency, where higher frequency corresponds to greater importance, thereby warranting higher allocated ranks, while never-routed experts receive zero ranks. However, infrequently routed experts may still encode non-negligible information essential for rare yet critical tasks (Chen et al., 2025; Gu et al., 2025).

One question naturally appears: ***how can we avoid overlooking information-rich but rarely activated experts?*** To identify such experts, we propose **effective rank** to measure the information density of each expert group.

The effective rank is computed from the normalized singular value spectrum and captures the intrinsic dimensionality of a matrix by measuring how many dominant singular components contribute meaningfully to its representation (Roy & Vetterli, 2007). Given a grouped expert matrix $\mathbf{W}_g \in \mathbb{R}^{kp \times d}(k = n/m)$ with singular values $\{\sigma_i\}_{i=1}^{r}$ obtained by singular value decomposition (SVD), we define the normalized spectral distribution as

$$p_i = \frac{\sigma_i^2}{\sum_{j=1}^{r} \sigma_j^2}, \tag{8}$$

where $r = \min(kp, d)$. The spectral entropy is given by

$$H(\mathbf{W}_g) = -\sum_{i=1}^{r} p_i \log p_i. \tag{9}$$

The effective rank of $W$ is defined as

$$\mathcal{R}_{eff}(\mathbf{W}_g) = \exp\big(H(\mathbf{W}_g)\big). \tag{10}$$

The effective rank indicates the number of ranks needed for a decomposed matrix to recover the information of the original matrix, and thus provides a measure of information density to guide rank allocation.

**Observation: Low-utilized experts can possess high information density.** Table 1 presents effective rank and routing frequency for expert groups in the first layer of Qwen3-30B-A3B-2507. The 128 experts are sorted by routing frequency and partitioned into 32 groups of four concatenated expert matrices each. We observe that the least activated expert groups may exhibit substantially higher effective ranks than more frequently activated groups. For instance, Group 30, despite a routing frequency of only 158 (lower than Groups 4, 9, 12, and 16), achieves an effective rank of 1049, demonstrating that low-utilized experts contain substantial information and should not be neglected.

| Expert Group ID | Routing Frequency | Effective Rank |
|---|---|---|
| Group 2 | 15111 | 1109 |
| Group 4 | 11421 | 929 |
| Group 9 | 8840 | 787 |
| Group 12 | 7235 | 799 |
| Group 16 | 5432 | 557 |
| Group 30 | 158 | 1049 |

*Table 1.* Routing frequency and effective rank.

### 3.3. Adaptive Rank Allocation via Experts Frequency and Information Density

Given the substantial variation in routing frequency across expert groups and the observation that infrequently routed experts may still possess high information density, we propose combining routing frequency with effective rank to guide adaptive rank allocation, balancing the importance indicated by activation patterns against the information density captured by effective rank. Based on Equations (8)-(10), the effective rank of the expert-group matrix $\mathbf{W}_g$ is computed as $\mathcal{R}_{eff}(\mathbf{W}_g)$. We then normalize the effective rank of each expert-group matrix into a probability distribution:

$$E_g = \frac{\mathcal{R}_{eff}(\mathbf{W}_g)}{\sum_{b=1}^{m} \mathcal{R}_{eff}(\mathbf{W}_b)} \tag{11}$$

Given the routing frequency and the normalized effective rank for each expert group, we fuse these two metrics using hyperparameter $\xi$:

$$C_g = \xi \cdot E_g + (1 - \xi) \cdot F_g \tag{12}$$

After fusion, the normalized importance score $C_g$ remains within $[0, 1]$, enabling direct use for rank allocation under a given target compression ratio. Specifically, given a target overall compression ratio, we first compute the total rank

budget $K_{\text{total}}$ determined by the parameter budget constraint. We then allocate the retained rank $K_g$ for each expert group proportionally to its fused importance score $C_g$, i.e.,

$$K_g = \max\left(1, \left\lfloor K_{\text{total}} \cdot \frac{C_g}{\sum_{b=1}^{m} C_g} \right\rfloor\right), \qquad (13)$$

where the lower bound ensures that each expert group retains minimal expressive capacity. The basis matrix for the $g$-th expert group is then denoted as $\mathbf{B}_g\big|_{K_g}$, and the corresponding coefficient matrix for expert $i$ in group $g$ is expressed as $\mathbf{A}_i\big|_{K_g}$. Accordingly, we reformulate Equation (4) as:

$$\hat{\mathbf{W}}_i \approx \mathbf{A}_i\big|_{K_g} \cdot \phi\left(\sum_{j=1}^{m} \alpha_{i,j} \mathbf{B}_j\big|_{K_j}\right) \qquad (14)$$

### 3.4. Residual Reconstruction

Finally, we begin to reconstruct the residual term $\mathbf{R}_i = \mathbf{W}_i - \hat{\mathbf{W}}_i$ for recovering more information discarded by the low-rank approximation in SVD-based compression: Specifically, we model the residual component using a low-dimensional trainable vector $\boldsymbol{\eta}$, enabling efficient reconstruction of $\mathbf{R}_i$ with minimal parameter overhead. Since $\boldsymbol{\eta}$ resides in a low-dimensional space, it must be embedded into the high-dimensional space of the expert weight matrix. This embedding should satisfy two key requirements: (i) parameter efficiency to minimize overhead, and (ii) structure preservation to maintain the correspondence between the low-dimensional representation and the original residual. In particular, the embedding should be topology-preserving to ensure that the intrinsic geometry of the residual is not distorted during reconstruction. Motivated by (Li et al., 2025a), we embed a low-dimensional trainable vector $\boldsymbol{\eta} \in \mathbb{R}^a$ into the original expert weight space using a sparse projection matrix $\mathbf{P} \in \mathbb{R}^{D \times a}$, reconstructing the residual as

$$\text{vec}(\hat{\mathbf{R}}_i) = \mathbf{P}\,\boldsymbol{\eta}_g, \qquad \hat{\mathbf{R}}_i = \text{reshape}(\mathbf{P}\,\boldsymbol{\eta}_g), \qquad (15)$$

where $D = kpd$ denotes the number of parameters of vectorized expert group matrix and $g$ is the expert group index.

We define $\mathbf{P}$ to have a one-hot structure per row: for each row $t \in \{1, \ldots, D\}$, we sample an index $\pi(t) \in \{1, \ldots, a\}$ uniformly at random and set

$$\mathbf{P}_{t,q} = \begin{cases} \frac{1}{\sqrt{n_q}} & q = \pi(t), \\ 0 & \text{otherwise,} \end{cases} \qquad (16)$$

where $n_q = \big|\{t : \pi(t) = q\}\big|$ is the frequency of $q$-th column to be sampled. This yields

$$\mathbf{P}^\top \mathbf{P} = \mathbf{I}_a, \qquad (17)$$

which makes the projection approximately geometry-preserving (Li et al., 2025a). In implementation, we do

---

**Algorithm 1** RFID-MoE

**Require:** MoE expert weights $\{\mathbf{W}_i\}_{i=1}^n$, calibration dataset $\mathcal{D}_{cal}$, target compression ratio $\rho$, fusion weight $\xi$, residual dimension $a$
**Ensure:** Compressed parameters for expert group $g$ $\{\mathbf{A}_i|_{K_g}, \mathbf{B}_g|_{K_g}, \boldsymbol{\eta}_g\}$
 1: Run $\mathcal{D}_{cal}$ on model and record routing counts $\{z_i\}_{i=1}^n$
 2: Compute expert activation frequencies $F_i = z_i / \sum_{b=1}^n z_b$
 3: Sort $F_i$ and corresponding experts in descending order
 4: Partition experts sequentially into $m$ groups, each containing $k = n/m$ experts
 5: **for** each expert group $g = 1, \ldots, m$ **do**
 6:     Concatenate expert matrices to form $\mathbf{W}_g$
 7:     Compute group activation frequency $F_g$
 8:     Perform SVD on $\mathbf{W}_g$ and obtain singular values $\{\sigma_i\}$
 9:     Compute normalized spectrum $p_i = \sigma_i^2 / \sum_j \sigma_j^2$
10:     Compute effective rank $\mathcal{R}_{eff}(\mathbf{W}_g) = \exp(-\sum_i p_i \log p_i)$
11: **end for**
12: Normalize effective ranks $E_g = \mathcal{R}_{eff}(\mathbf{W}_g) / \sum_b \mathcal{R}_{eff}(\mathbf{W}_b)$
13: Fuse metrics $C_g = \xi E_g + (1 - \xi) F_g$
14: Compute total rank budget $K_{\text{total}}$ from $\rho$
15: Allocate group ranks $K_g = \max\left(1, \left\lfloor K_{\text{total}} \cdot \frac{C_g}{\sum_b C_b} \right\rfloor\right)$
16: **for** each expert group $g$ **do**
17:     Perform truncated SVD on $\mathbf{W}_g$ with rank $K_g$ to obtain $\mathbf{B}_g|_{K_g}$ and $\{\mathbf{A}_i|_{K_g}\}$
18:     Initialize low-dimensional residual vector $\boldsymbol{\eta}_g \in \mathbb{R}^a$
19: **end for**
20: Construct sparse projection matrix $\mathbf{P}$ and share it across groups
21: Reconstruct residuals as $\hat{\mathbf{R}}_i = \text{reshape}(\mathbf{P}\boldsymbol{\eta}_g)$
22: **Return** $\{\mathbf{A}_i|_{K_g}, \mathbf{B}_g|_{K_g}, \boldsymbol{\eta}_g\}$

---

not need to store the entire matrix $\mathbf{P}$; we store only the index $\pi(t)$ and the scaling factors $1/\sqrt{n_{\pi(t)}}$ to recover $\mathbf{P}$.

We construct the projection matrix $\mathbf{P}$ once and share it across all expert groups, thereby constraining all residuals to lie within the same embedding subspace. This design offers two key advantages: (1) Since $\mathbf{P}$ is an isometric mapping, it preserves the geometric structure of the optimization landscape, ensuring that the low-dimensional residual representation is not distorted during projection (Li et al., 2018); (2) Sharing $\mathbf{P}$ across expert groups enforces a unified residual subspace, which promotes structural alignment and facilitates parameter sharing among experts. This shared embedding not only reduces the total parameter count, but also improves training stability (Li et al., 2025a; Wang et al., 2024). We provide the effectiveness analysis of our residual reconstruction in Appendix C.

*Table 2.* Performance of Our method on Qwen3-30B-A3B-2507, DeepSeekMoE-16B-Base, Deepseek-v2-Lite-Chat and Qwen2-57B-A14B. Perplexity ($\downarrow$) is reported for language modeling datasets and accuracy ($\uparrow$) for common sense reasoning datasets.

| Ratio | Method | WikiText-2 ($\downarrow$) | PTB ($\downarrow$) | C4 ($\downarrow$) | Open. ($\uparrow$) | ARC-e ($\uparrow$) | WinoG. ($\uparrow$) | HellaS. ($\uparrow$) | PIQA ($\uparrow$) | MathQA ($\uparrow$) | Average ($\uparrow$) |
|---|---|---|---|---|---|---|---|---|---|---|---|
| | | | | | **Qwen3-30B-A3B-2507** | | | | | | |
| 0% | Original | 7.32 | 12.41 | 12.49 | 0.42 | 0.78 | 0.70 | 0.69 | 0.79 | 0.58 | 0.66 |
| 20% | NAEE | 8.95 | 14.18 | 13.77 | 0.42 | 0.76 | 0.69 | 0.68 | 0.78 | 0.51 | 0.64 |
| | $D^2$-MoE | 9.12 | 17.64 | 18.28 | 0.41 | 0.73 | 0.66 | 0.64 | 0.76 | 0.49 | 0.62 |
| | RS-MoE | 8.87 | 13.93 | 13.36 | 0.42 | 0.77 | 0.67 | 0.68 | 0.79 | 0.53 | 0.64 |
| | MoBE | 7.50 | 12.73 | 12.69 | **0.46** | **0.85** | 0.73 | **0.79** | 0.81 | 0.63 | 0.71 |
| | **RFID-MoE** | **7.37** | **12.53** | **12.57** | 0.45 | **0.85** | **0.74** | **0.79** | **0.82** | **0.64** | **0.72** |
| 40% | NAEE | 10.07 | 15.28 | 14.93 | 0.40 | 0.70 | 0.65 | 0.63 | 0.75 | 0.44 | 0.60 |
| | $D^2$-MoE | 14.47 | 26.58 | 21.72 | 0.37 | 0.67 | 0.62 | 0.59 | 0.72 | 0.40 | 0.56 |
| | RS-MoE | 9.48 | 15.10 | 15.05 | 0.39 | 0.71 | 0.66 | 0.65 | 0.77 | 0.44 | 0.60 |
| | MoBE | 8.17 | 13.99 | 13.92 | 0.44 | 0.83 | 0.71 | 0.75 | **0.80** | 0.62 | 0.69 |
| | **RFID-MoE** | **7.59** | **12.81** | **13.21** | **0.46** | **0.85** | 0.73 | 0.77 | **0.80** | 0.63 | 0.71 |
| 60% | NAEE | 13.76 | 19.22 | 20.01 | 0.34 | 0.65 | 0.60 | 0.58 | 0.70 | 0.35 | 0.54 |
| | $D^2$-MoE | 21.76 | 38.84 | 36.55 | 0.29 | 0.60 | 0.58 | 0.52 | 0.65 | 0.33 | 0.50 |
| | RS-MoE | 13.56 | 20.17 | 20.12 | 0.34 | 0.63 | 0.61 | 0.60 | 0.71 | 0.39 | 0.55 |
| | MoBE | 13.96 | 24.93 | 24.40 | **0.40** | 0.78 | 0.67 | 0.58 | 0.74 | 0.51 | 0.61 |
| | **RFID-MoE** | **9.57** | **16.92** | **17.96** | 0.39 | **0.80** | **0.71** | **0.66** | **0.77** | **0.52** | **0.64** |
| | | | | | **DeepSeekMoE-16B-Base** | | | | | | |
| 0% | Original | 6.51 | 9.74 | 10.20 | 0.45 | 0.70 | 0.70 | 0.77 | 0.80 | 0.31 | 0.62 |
| 40% | NAEE | 8.55 | 14.47 | 17.98 | 0.23 | 0.67 | 0.66 | 0.41 | 0.69 | 0.26 | 0.49 |
| | MoE-I$^2$ | 9.73 | 15.75 | 19.75 | 0.23 | 0.64 | 0.66 | 0.41 | 0.68 | 0.26 | 0.48 |
| | $D^2$-MoE | 7.93 | 14.07 | 15.18 | 0.26 | 0.69 | 0.65 | 0.45 | 0.72 | 0.28 | 0.51 |
| | RS-MoE | 8.15 | 13.26 | 14.93 | 0.28 | 0.67 | 0.68 | 0.48 | 0.73 | 0.28 | 0.52 |
| | MoBE | 6.94 | 10.36 | 10.33 | 0.42 | 0.69 | **0.70** | 0.74 | 0.79 | 0.30 | 0.61 |
| | **RFID-MoE** | **6.78** | **10.18** | **10.20** | **0.43** | **0.70** | 0.69 | **0.76** | **0.80** | **0.31** | **0.62** |
| 60% | NAEE | 23.20 | 49.89 | 48.63 | 0.17 | 0.49 | 0.58 | 0.33 | 0.61 | 0.23 | 0.34 |
| | MoE-I$^2$ | 15.83 | 32.20 | 38.60 | 0.17 | 0.48 | 0.58 | 0.32 | 0.61 | 0.22 | 0.40 |
| | $D^2$-MoE | 11.67 | 27.73 | 27.63 | 0.21 | 0.54 | 0.61 | 0.35 | 0.63 | 0.24 | 0.43 |
| | RS-MoE | 9.95 | 18.29 | 22.52 | 0.26 | 0.59 | 0.65 | 0.40 | 0.68 | 0.26 | 0.47 |
| | MoBE | 8.33 | 11.90 | 12.34 | 0.40 | 0.67 | 0.68 | 0.69 | 0.77 | 0.29 | 0.58 |
| | **RFID-MoE** | **7.95** | **11.54** | **12.06** | **0.41** | **0.68** | **0.69** | **0.71** | **0.79** | **0.30** | **0.60** |
| | | | | | **Qwen2-57B-A14B** | | | | | | |
| 0% | Original | 5.12 | 9.18 | 8.86 | 0.45 | 0.71 | 0.74 | 0.85 | 0.83 | 0.39 | 0.67 |
| 40% | NAEE | 6.81 | 11.34 | 11.57 | 0.31 | 0.73 | 0.73 | 0.55 | 0.76 | 0.36 | 0.57 |
| | MoE-I$^2$ | 24.90 | 77.05 | 22.50 | 0.26 | 0.70 | 0.46 | 0.71 | 0.75 | 0.30 | 0.53 |
| | $D^2$-MoE | 8.19 | 11.23 | 12.70 | 0.33 | 0.75 | 0.75 | 0.61 | 0.79 | 0.36 | 0.60 |
| | MoBE | 6.27 | 9.58 | 9.27 | 0.45 | 0.70 | 0.75 | 0.82 | 0.82 | **0.40** | 0.66 |
| | **RFID-MoE** | **6.14** | **9.46** | **9.16** | **0.46** | **0.71** | **0.76** | **0.84** | **0.83** | 0.39 | **0.67** |
| | | | | | **Qwen1.5-MoE-A2.7B** | | | | | | |
| 0% | Original | 7.30 | 11.49 | 10.18 | 0.45 | 0.78 | 0.71 | 0.81 | 0.80 | 0.39 | 0.66 |
| 40% | MoBE | 7.81 | 12.16 | 11.03 | 0.41 | 0.65 | **0.69** | 0.75 | 0.79 | **0.36** | 0.61 |
| | **RFID-MoE** | **7.63** | **11.98** | **10.90** | **0.42** | **0.68** | 0.67 | **0.76** | **0.80** | **0.36** | **0.62** |
| 60% | MoBE | 9.20 | 13.82 | 13.20 | **0.41** | 0.66 | 0.68 | 0.69 | 0.77 | 0.33 | 0.59 |
| | **RFID-MoE** | **8.92** | **13.66** | **12.99** | 0.39 | **0.67** | **0.69** | **0.71** | **0.78** | **0.34** | **0.60** |

Finally, Equation (14) can be reformulated as:

$$\hat{\mathbf{W}}_i \approx \mathbf{A}_i\big|_{K_g} \cdot \phi\left(\sum_{j=1}^{m} \alpha_{i,j}\mathbf{B}_j\big|_{K_j}\right) + reshape_i(\mathbf{P}\boldsymbol{\eta}_\mathbf{g}) \quad (18)$$

After the lightweight residual reconstruction training defined in Equation (5), we obtain the final parameters for each expert: the expert-specific coefficient matrices $\mathbf{A}_i\big|_{K_g}$, the group-shared basis matrices $\mathbf{B}_g\big|_{K_g}$, and the low-dimensional residual vectors $\{\boldsymbol{\eta}_g\}$. The overall compression procedure is detailed in Algorithm 1.

## 4. Experiments

**Models.** We evaluate RFID-MoE on various MoE-based LLMs, including: Qwen3-30B-A3B-2507, DeepSeek-MoE-16B-base, Qwen2-57B-A14B, and Deepseek-v2-Lite-Chat.

These models exhibit different expert-level designs: Qwen3-30B-A3B-2507 with 128 experts per layer, while DeepSeek-MoE-16B-base, Qwen2-57B-A14B, and Deepseek-v2-Lite-Chat with up to 64 experts per layer.

**Baselines.** We compare our method against various baselines: NAEE (Lu et al., 2024) combines post-training expert pruning with dynamic expert skipping to reduce computational costs during inference; MoE-I$^2$ (Yang et al., 2024) integrates inter-expert pruning and intra-expert low-rank decomposition to compress MoE models with minimal performance degradation; RS-MoE (Anonymous, 2025) proposes a collaborative compression framework that decomposes expert parameters into low-rank shared representations and sparse expert-specific residuals; D$^2$-MoE (Gu et al., 2025) splits expert weights into shared and unique components to achieve efficient compression; MoBE (Chen et al., 2025)

*Table 3.* Comparison of PPL ($\downarrow$) on WikiText-2, PTB, and C4 under different calibration datasets for frequency calculation. The best results are highlighted in bold.

| Method | Calibration | WikiText-2 | PTB | C4 |
|---|---|---|---|---|
| $D^2$-MoE | | 21.76 | 38.84 | 36.55 |
| RS-MoE | – | 13.56 | 20.17 | 20.12 |
| MoBE | | 13.96 | 24.93 | 24.40 |
| **RFID-MoE (Ours)** | WikiText-2 | **9.57** | 16.92 | 17.96 |
| | C4 | 10.51 | **13.70** | **16.85** |

employs adaptive basis sharing across different experts for efficient MoE model compression.

**Datasets.** We evaluate RFID-MoE on two types of benchmarks. For language modeling, we use WikiText-2 (Merity et al., 2016), PTB (Marcus et al., 1993), and C4 (Raffel et al., 2020). For reasoning, we assess performance on six common-sense and reasoning tasks: OpenBookQA (Mihaylov et al., 2018), WinoGrande (Sakaguchi et al., 2021), HellaSwag (Zellers et al., 2019), PIQA (Bisk et al., 2020), MathQA (Amini et al., 2019), ARC-e (Clark et al., 2018). All evaluations are conducted in a zero-shot setting using the LM Evaluation Harness framework (Gao et al., 2024).

**Implementation Details.** During the expert activation frequency extraction stage, we collect routing statistics using 1,024 samples from WikiText-2. We also report results using C4 as calibration data in the ablation studies. Following the frequency analysis, we compress the up and gate matrices of all experts across the model. For basis construction, we group every four expert matrices and apply SVD-based low-rank decomposition. Following (Li et al., 2025a), the parameter size of the vector $\eta$ is set to 3% of the original parameter count, with the parameter budgets of the main components $A$ and $B$ reduced accordingly. We employ SiLU activation function in our main result experiments. All experiments are conducted on 8 NVIDIA A100 GPUs.

### 4.1. Main Results

**Language Modeling Performance.** Table 2 reports perplexity (PPL) results across different compression ratios and MoE baselines. Overall, RFID-MoE consistently achieves the lowest or near-lowest PPL at each compression ratio, demonstrating superior language modeling capability compared to existing baselines. The advantage becomes more substantial at higher compression ratios. For example, on Qwen3-30B-A3B-2507 at 60% compression ratio, RFID-MoE achieves a PPL of 16.92 on PTB, representing an improvement of approximately 8.0 points over the best competing baseline (i.e., MoBE). Similar improvements are observed on DeepSeekMoE-16B-Base, where RFID-MoE consistently yields lower PPL at both 40% and 60% compression ratios, with up to 0.4 PPL reduction on PTB. These results demonstrate that RFID-MoE effectively preserves language modeling capacity under aggressive compression.

*Table 4.* Comparison of PPL ($\downarrow$) on WikiText-2 and PTB under different compression ratios and $\xi$ values.

| Compression ratio | Method | $\xi$ | WikiText-2 | PTB |
|---|---|---|---|---|
| 20% | $D^2$-MoE | – | 9.12 | 17.64 |
| | RS-MoE | | 8.87 | 13.93 |
| | MoBE | | 7.50 | 12.73 |
| | **RFID-MoE (Ours)** | 0.9 | 7.40 | 12.58 |
| | | 0.8 | **7.37** | **12.53** |
| | | 0.7 | 7.39 | 12.55 |
| | | 0.5 | 7.42 | 12.54 |
| 40% | $D^2$-MoE | – | 14.47 | 26.58 |
| | RS-MoE | | 9.48 | 15.10 |
| | MoBE | | 8.17 | 13.99 |
| | **RFID-MoE (Ours)** | 0.9 | 7.82 | 13.09 |
| | | 0.8 | 7.68 | 12.88 |
| | | 0.7 | **7.59** | **12.81** |
| | | 0.5 | 7.65 | 12.86 |
| 50% | $D^2$-MoE | – | 21.76 | 38.84 |
| | RS-MoE | | 13.56 | 20.17 |
| | MoBE | | 13.96 | 24.93 |
| | **RFID-MoE (Ours)** | 0.8 | 8.43 | 14.57 |
| | | 0.7 | **8.02** | **13.53** |
| | | 0.5 | 8.23 | 14.03 |

**Zero-shot Accuracy on Reasoning Benchmarks.** RFID-MoE demonstrates consistent gains on zero-shot tasks. On Qwen3-30B-A3B-2507 at 60% compression ratio, RFID-MoE outperforms MoBE and RS-MoE on nearly all tasks, achieving 0.66 on HellaSwag (approximately 8% improvement) and 0.71 on WinoGrande (about 4% higher). Across different compression ratios, RFID-MoE consistently delivers the best or near-best accuracy on ARC-e, PIQA, and HellaSwag. Similar trends appear on DeepSeekMoE-16B-Base, where RFID-MoE achieves the highest average accuracy at both 40% and 60% compression ratios. At 60% compression ratio, RFID-MoE obtains 2%-3% higher accuracy on HellaSwag and PIQA compared to MoBE.

### 4.2. Ablation Study

**Different Calibration Datasets for Routing Frequency Calculation.** Table 3 presents RFID-MoE results on routing frequency under different calibration settings. RFID-MoE consistently outperforms the state-of-the-art baselines. For example, at 60% compression ratio using C4 and WikiText-2 as calibration data to estimate routing frequency, RFID-MoE achieves PPL of 16.85 and 17.96 on C4, achieving over 7.55 and 6.44 lower (better) PPL than baselines, respectively.

**Impact of $\xi$.** Table 4 reports the compression performance on WikiText-2 and PTB under different fusion hyperparameter $\xi$ values. The optimal $\xi$ values concentrate around $0.7 \sim 0.8$, consistent with our analysis in Section 3.2. When $\xi$ is too large, rank allocation relies predominantly on effective rank and disregards activation patterns. Conversely, when $\xi$ is too small, allocation becomes dominated by routing frequency, assigning insufficient rank to low-frequency

*Table 5.* Language modeling performance of our method on Deepseek-V2-Lite-Chat and Ling-mini-2.0 models. Perplexity (↓) is reported on WikiText-2, PTB, and C4.

| Ratio | Method | WikiText-2 | PTB | C4 |
|---|---|---|---|---|
| | **Deepseek-V2-Lite-Chat** | | | |
| | Original | 8.01 | 11.79 | 11.85 |
| 40% | MoBE | 8.43 | 12.17 | 12.67 |
| | **RFID-MoE (Ours)** | **8.24** | **11.90** | **12.48** |
| 50% | MoBE | 9.06 | 12.94 | 13.48 |
| | **RFID-MoE (Ours)** | **8.76** | **12.52** | **13.29** |
| | **Ling-mini-2.0** | | | |
| | Original | 14.33 | 23.27 | 23.26 |
| 40% | MoBE | 14.53 | 25.48 | 23.81 |
| | **RFID-MoE (Ours)** | **14.37** | **24.18** | **23.35** |

*Table 6.* Effect of Residual Vector under 50% Compression on Qwen3-30B-A3B-2507. Lower perplexity (PPL) indicates better performance.

| Method | WikiText PPL ↓ | PTB PPL ↓ |
|---|---|---|
| MoBE | 10.00 | 17.41 |
| **RFID-MoE (W/O residual)** | 8.91 | 15.06 |
| **RFID-MoE (W/ residual)** | **8.02** | **13.53** |

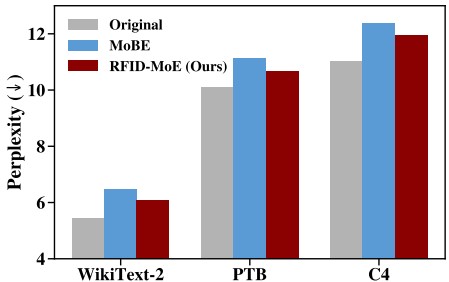

*Figure 3.* Language modeling performance of our method on Qwen3-235B-A22B-2507. Perplexity (↓) is reported on WikiText-2, PTB, and C4.

expert groups. This degrades to a pruning-based approach, compromising the model's capacity to handle specialized tasks that require these infrequently activated experts.

**Experiments on Large Model.** We evaluate the effectiveness of RFID-MoE on a larger-scale MoE model Qwen3-235B-A22B-2507, as shown in Figure 3. We observe that RFID-MoE consistently outperforms MoBE. For example, on PTB, RFID-MoE achieves a PPL of 10.67, compared to 11.13 for MoBE, yielding an improvement of 0.5 PPL. Moreover, this result is only 0.5 higher than the original model's PPL of 10.11, indicating that RFID-MoE preserves most of the original model's language modeling capability even at high compression ratios.

**Experiments on Lightweight Model.** We evaluate RFID-MoE on lightweight models in Table 5. DeepSeek-V2-Lite-Chat and Ling-mini-2.0 are both MoE models with approximately 15B total parameters, while activating only 2.4B and 1.4B parameters during inference, respectively. As shown

*Table 7.* Ablation on group size under 40% compression using Qwen3-30B-A3B-2507 with RFID-MoE. The product of group size and the number of groups is fixed to the number of experts per layer (128). Lower perplexity (PPL) indicates better performance.

| Group Size ($k$) | Groups ($m$) | PTB PPL ↓ | WikiText PPL ↓ |
|---|---|---|---|
| 2 | 64 | 14.04 | 8.55 |
| 4 | 32 | **12.81** | **7.59** |
| 8 | 16 | 12.85 | **7.59** |

*Table 8.* Comparison of zero-shot accuracy under 50% compression on Qwen3-30B-A3B-2507. Higher accuracy indicates better performance.

| Method | MMLU | WinoG. | HellaSwag | ARC_c |
|---|---|---|---|---|
| REAP | 0.50 | 0.58 | 0.48 | 0.35 |
| MoBE | 0.73 | 0.72 | 0.68 | 0.59 |
| **RFID-MoE (Ours)** | **0.75** | **0.73** | **0.74** | **0.60** |

in the table, our method consistently outperforms the MoBE on these models. For example, at 50% compression ratio, RFID-MoE achieves around 0.4 lower PPL than MoBE on the PTB dataset for DeepSeek-V2-Lite-Chat, and around 0.3 lower PPL for Ling-mini-2.0 of 40% compression ratio. These results demonstrate that RFID-MoE remains effective even on lightweight MoE models with sparse activation.

**Effect of Residual Reconstruction.** We further conduct experiments to decouple the effects of adaptive rank allocation and residual reconstruction, as summarized in Table 6. Under 50% compression on Qwen3-30B-A3B-2507, adaptive rank allocation alone reduces PTB PPL from 17.41 to 15.06 and WikiText-2 PPL from 10.00 to 8.91. Further incorporating the residual vector consistently improves performance, achieving 13.53 PTB PPL and 8.02 WikiText-2 PPL.

**Impact of $k$ and $m$.** We further provide an ablation study on group size and the number of groups under 40% compression using Qwen3-30B-A3B-2507, as shown in Table 7. The product of group size and the number of groups is fixed to the number of experts per layer (128). We observe that increasing the group size from 2 to 4 significantly improves performance, while further increasing it to 8 brings marginal gains, suggesting that moderate group sizes achieve a better trade-off between expert sharing and reconstruction quality.

**Comparison with SOTA expert pruning method.** Table 8 presents the accuracy of REAP, a state-of-the-art expert pruning method, MoBE, and RFID-MoE across four datasets using Qwen3-30B-A3B-2507 under 50% compression ratios. REAP consistently underperforms both MoBE and RFID-MoE, while our method achieves up to a 6% improvement over MoBE on HellaSwag at 50% compression. These results further support our discussion in Section 1 that relying solely on routing frequency for expert pruning can severely degrade performance on specialized tasks where infrequently-activated experts encode critical domain-specific knowledge.

*Table 9.* Comparison of PPL (↓) on WikiText-2 and PTB under different dimension of $\eta$ at 50% compression.

| Method | $\eta$ | WikiText-2 | PTB |
|---|---|---|---|
| MoBE | – | 10.00 | 17.41 |
| **RFID-MoE (Ours)** | 0% | 8.91 | 15.06 |
| | 1% | 8.52 | 14.54 |
| | 3% | **8.02** | **13.53** |
| | 5% | 9.90 | 16.52 |

**Impact of dimension for $\eta$.** Table 9 presents the effect of different residual vector sizes under 50% compression, where the $\eta$ column denotes the parameter ratio of the residual vector relative to the original expert parameters. We observe that introducing a small residual vector consistently improves performance over the MoBE baseline and the variant without residual reconstruction ($\eta = 0\%$). In particular, setting $\eta = 3\%$ achieves the best performance on both WikiText-2 and PTB, while a larger residual size ($\eta = 5\%$) leads to noticeable degradation.

## 5. Related Work

**Mixture-of-Experts.** The MoE architecture has emerged as a dominant paradigm for scaling AI models, enabling efficient model capacity expansion through sparse gating mechanisms that activate only subsets of experts per input (Clark et al., 2022; Jiang et al., 2024; Dai et al., 2024; Rajbhandari et al., 2022). This efficiency has driven widespread MoE adoption in LLMs, with recent research refining expert structures, router designs, and training methods (Zhou et al., 2022; Shen et al., 2025a; Yang et al., 2025b; Shen et al., 2025d; Liu et al., 2023). Despite these advances, MoE architectures face deployment challenges: expert replication across devices causes inflated parameter budgets and memory overheads, while distributed expert computation introduces communication costs that offset theoretical computational savings. Addressing these limitations while preserving MoE's scaling advantages remains an open challenge.

**MoE-based LLM Compression.** Three categories of MoE-based LLM Compression approaches exist: pruning eliminates non-critical expert paths (Xie et al., 2024; Chen et al., 2026; Lu et al., 2024; Yang et al., 2024), merging consolidates experts through clustering and fusion (He et al., 2023; Miao et al., 2025; Li et al., 2023; Liu et al., 2024b), and decomposition approximates weights via matrix factorization. Recently, decomposition-based methods have gained prominence by preserving expert architectures. $D^2$-MoE (Gu et al., 2025) and MoBE (Chen et al., 2025) approximate expert weights via low-rank factorization, preserving expert architectures while achieving high performance (Chen et al., 2025; Li et al., 2025c; Gu et al., 2025). Unlike pruning and merging, these methods avoid explicit expert reduction, achieving superior performance across benchmarks (Chen

et al., 2025; Li et al., 2025c; Gu et al., 2025). However, existing methods ignore the extreme imbalance in expert routing frequencies across experts, causing uniform or static rank allocation to yield suboptimal results (Chen et al., 2025; Li et al.; He et al., 2024; Leyang et al., 2025; Li et al., 2025b).

## 6. Conclusion

In this paper, we introduce RFID-MoE, a compression framework tailored for MoE LLMs. Unlike existing work using uniform rank or static weight property for compression, RFID-MoE explicitly addresses expert heterogeneity by dynamically allocating ranks based on routing frequency and information density. Furthermore, we propose a parameter-efficient residual reconstruction method using sparse orthogonal projection, which is effective to recover critical information discarded by low-rank approximation. Extensive experiments demonstrate that RFID-MoE consistently outperforms state-of-the-art baselines in both perplexity and reasoning accuracy, offering a practical solution for deploying massive MoE models on resource-constrained devices.

## Limitations

Although RFID-MoE demonstrates strong performance on language-based MoE models, it remains unclear whether the proposed method can be effectively extended to VLM-based MoE architectures, where expert behaviors and multimodal representations may differ significantly from pure language models. In addition, our work mainly focuses on the compression perspective, while the system-level acceleration of grouped SVD-based MoE models remains underexplored. Designing efficient inference kernels and optimized system support for grouped low-rank computation is an important direction for future work.

## Impact Statement

This paper presents work whose goal is to advance the field of AI and Machine Learning. There are many potential societal consequences of our work, none which we feel must be specifically highlighted here. Our method focuses on improving the efficiency of large-scale MoE models, which may help reduce computational and energy costs for future AI systems.

## Acknowledgements

This research used resources of the Argonne Leadership Computing Facility, a U.S. Department of Energy Office of Science User Facility, operated under contract DE-AC02-06CH11357. This work was in part supported by the NSF OAC-2348465, NSF 2527416, 2534241, and 2523997. Any opinions, findings, conclusions, and recommendations expressed in this material are those of the authors and do not necessarily reflect the views of the funding agencies.

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

# Appendix

## A. Convergence Comparison

Figure 4 compares the training scaled loss, defined as the training error normalized by the standard deviation of the target parameter values, for the gate and up matrices at layer 47 of Qwen-3-30B-A3B-2507. As shown in both subfigures, RFID-MoE consistently achieves faster convergence and lower final loss compared to MoBE. The improvement is particularly evident in the early stages of training, where RFID-MoE rapidly reduces the scaled loss, indicating more efficient and stable optimization. This result demonstrates that by integrating routing frequency with effective rank for adaptive rank allocation, our method not only improves overall model performance but also enhances convergence speed during the compression and reconstruction stages.

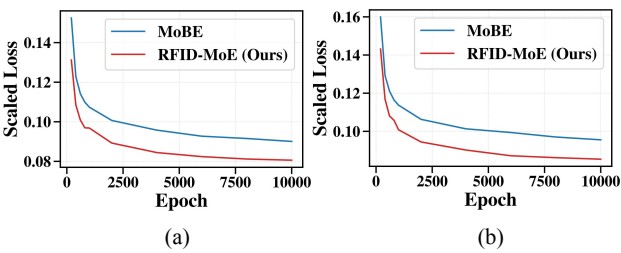

(a)  (b)

*Figure 4.* Training scaled loss comparison between MoBE and RFID-MoE (Ours) for 47-th layer of Qwen3-30B-A3B-2507. (a) Gate projection matrix. (b) Up projection matrix

## B. Theoretical Analysis

In this section, we provide a rigorous mathematical derivation for the construction of the sparse projection matrix $P$ and analyze why the proposed residual reconstruction method is effective. Specifically, we prove that our construction of $P$ guarantees strict isometry from the low-dimensional parameter space to the high-dimensional weight space, ensuring optimization stability. Furthermore, we provide a theoretical justification for the effectiveness of residual reconstruction based on the intrinsic dimension hypothesis.

### B.1. Construction and Isometry of Projection Matrix $P$

**Definition 1 (Sparse Normalized Projection Matrix $P$).** Let $D$ be the dimension of the vectorized expert residual $R \in \mathbb{R}^D$, and $a$ be the dimension of the trainable low-rank vector $\eta \in \mathbb{R}^a$, with $a \ll D$. We define a mapping function $\pi : \{1, \ldots, D\} \to \{1, \ldots, a\}$ where each index $t \in \{1, \ldots, D\}$ is assigned to a latent dimension index $\pi(t)$ sampled uniformly at random. The projection matrix $P \in \mathbb{R}^{D \times a}$ is constructed as follows:

$$P_{t,q} = \begin{cases} \frac{1}{\sqrt{n_q}} & \text{if } q = \pi(t), \\ 0 & \text{otherwise,} \end{cases} \quad (19)$$

where $n_q = |\{t \mid \pi(t) = q\}|$ denotes the frequency count of the $q$-th latent dimension being selected.

**Proposition 1 (Column Orthonormality).** The matrix $P$ constructed in Definition 1 satisfies the orthogonality condition $P^T P = I_a$, where $I_a$ is the $a \times a$ identity matrix.

*Proof.* Let $G = P^T P$ be the Gram matrix of $P$. The element $G_{ij}$ is given by the inner product of the $i$-th and $j$-th columns of $P$:

$$G_{ij} = \sum_{t=1}^{D} P_{t,i} P_{t,j}. \quad (20)$$

**Case 1: Off-diagonal elements ($i \neq j$).** By the definition of $\pi(t)$, for any row $t$, there is a unique column index $q = \pi(t)$ such that $P_{t,q} \neq 0$. This implies that for any $t$, $P_{t,i}$ and $P_{t,j}$ cannot be simultaneously non-zero if $i \neq j$. Specifically:

- If $\pi(t) = i$, then $P_{t,i} \neq 0$ and $P_{t,j} = 0$.
- If $\pi(t) = j$, then $P_{t,j} \neq 0$ and $P_{t,i} = 0$.
- Otherwise, both are zero.

Thus, the product $P_{t,i} P_{t,j} = 0$ for all $t$. Consequently,

$$G_{ij} = \sum_{t=1}^{D} 0 = 0, \quad \forall i \neq j. \quad (21)$$

**Case 2: Diagonal elements ($i = j$).** Consider the diagonal element $G_{ii}$:

$$G_{ii} = \sum_{t=1}^{D} (P_{t,i})^2. \quad (22)$$

The term $P_{t,i}$ is non-zero if and only if $\pi(t) = i$. Let $S_i = \{t \mid \pi(t) = i\}$ be the set of row indices mapped to column $i$. By definition, $|S_i| = n_i$. Substituting Eq. (19):

$$G_{ii} = \sum_{t \in S_i} \left(\frac{1}{\sqrt{n_i}}\right)^2 = \sum_{t \in S_i} \frac{1}{n_i} = n_i \cdot \frac{1}{n_i} = 1. \quad (23)$$

Combining both cases, we have $G = I_a$. $\qquad \square$

**Corollary 1 (Isometry Property).** The mapping induced by $P$ is an isometry. For any vector $\eta \in \mathbb{R}^a$, the Euclidean norm is preserved in the high-dimensional space:

$$\|P\eta\|_2^2 = (P\eta)^T (P\eta) = \eta^T (P^T P)\eta = \eta^T I_a \eta = \|\eta\|_2^2. \quad (24)$$

**Remark 1 (Optimization Stability).** The isometry property is crucial for training stability. It ensures that the gradient flow is not distorted. Specifically, considering the loss function $\mathcal{L}$, the gradient with respect to $\eta$ is $\nabla_\eta \mathcal{L} = P^T \nabla_{\hat{R}} \mathcal{L}$. Since $P$ has orthonormal columns, the condition number of the transformation is 1, preventing the exploding or vanishing gradient problems often associated with random projections where column norms are not strictly controlled.

# C. Theoretical Analysis of Residual Reconstruction Effectiveness

In this section, we provide a rigorous theoretical justification for the effectiveness of the proposed residual reconstruction method. We demonstrate that, under the hypothesis of a low intrinsic dimension of the optimization landscape, a sparse random projection $P$ can recover a solution that is $\epsilon$-close to the optimal residual with high probability.

## C.1. Problem Formulation

Let $W^* \in \mathbb{R}^D$ be the optimal weight matrix (vectorized) for an expert, and let $\hat{W}_{SVD}$ be the low-rank approximation obtained via SVD. The target residual is defined as $R^* = W^* - \hat{W}_{SVD}$. Our method attempts to approximate $R^*$ using a vector $P\eta$, where $P \in \mathbb{R}^{D \times a}$ is the proposed sparse orthogonal projection matrix ($a \ll D$) and $\eta \in \mathbb{R}^a$ is a trainable vector. The reconstruction error is given by:

$$\mathcal{E} = \min_{\eta \in \mathbb{R}^a} \|R^* - P\eta\|_2^2. \qquad (25)$$

## C.2. Strict Isometry of the Projection $P$

First, we establish the geometric property of $P$. As proven in Proposition 1, $P$ satisfies $P^T P = I_a$. This implies that $P$ is an isometric embedding from $\mathbb{R}^a$ to a subspace $\mathcal{S} \subset \mathbb{R}^D$ of dimension $a$. Consequently, for any gradients $\nabla_\eta \mathcal{L}$ in the low-dimensional space, the update dynamics are strictly preserved in the high-dimensional space without scaling distortion:

$$\|\nabla_\eta \mathcal{L}\|_2 = \|P^T \nabla_{\hat{R}} \mathcal{L}\|_2. \qquad (26)$$

## C.3. Approximation Bound via Intrinsic Dimension

We rely on the *Intrinsic Dimension* hypothesis (Li et al., 2018), which posits that the solution space of large language models is not a single point but a low-dimensional manifold embedded in $\mathbb{R}^D$.

**Assumption 1 (Manifold of Solution).** For a given error tolerance $\epsilon > 0$, the set of valid residuals that maintain model performance, denoted as $\mathcal{M}_\epsilon = \{R \in \mathbb{R}^D \mid \mathcal{L}(\hat{W}_{SVD}+R) \leq \mathcal{L}(W^*)+\epsilon\}$, contains a $d_{int}$-dimensional linear subspace $\mathcal{V} \subset \mathbb{R}^D$, where $d_{int} \ll D$.

**Theorem 1 (Existence of $\epsilon$-Approximate Solution).** Let $\mathcal{V}$ be the $d_{int}$-dimensional solution subspace. Let $\mathcal{S} =$ Range$(P)$ be the $a$-dimensional random subspace spanned by the columns of $P$. If the projection dimension $a$ satisfies:

$$a \geq d_{int} + 2\log(1/\delta) + C, \qquad (27)$$

where $\delta \in (0,1)$ is the failure probability and $C$ is a constant, then with probability at least $1 - \delta$, the intersection of the random subspace $\mathcal{S}$ and the solution subspace $\mathcal{V}$ is

non-trivial, or sufficiently close such that:

$$\exists \eta^* \in \mathbb{R}^a \quad \text{s.t.} \quad \|P\eta^* - \text{proj}_{\mathcal{V}}(R^*)\|_2^2 \leq \epsilon. \qquad (28)$$

*Proof.* The proof utilizes the geometry of Grassmannian manifolds and the concentration of measure.

**Decomposition:** Any target residual $R^*$ can be decomposed into a component within the intrinsic solution subspace $R_{\mathcal{V}} \in \mathcal{V}$ and an orthogonal noise component $R_\perp$. The SVD-based initialization ensures that $\|R_\perp\|$ is already minimized (capturing high-energy components).

**Subspace Intersection Probability:** We consider the random subspace $\mathcal{S}$ generated by $P$. The probability that a random $a$-dimensional subspace captures a significant fraction of the energy of an arbitrary fixed vector is low. However, we are not fitting a fixed vector; we are searching for *any* vector in the intersection $\mathcal{S} \cap \mathcal{V}$ that satisfies the loss constraint.

**Gordon's Escape Theorem:** Borrowing from results in compressed sensing (Gordon's Escape through a Mesh), if the dimension $a$ exceeds the intrinsic Gaussian width $w(\mathcal{V})$ of the solution set which is proportional to $d_{int}$, the random subspace $\mathcal{S}$ will intersect the cone of descent directions with high probability.

Since $P$ is a random orthogonal projection (a specific instance of Johnson-Lindenstrauss embedding), it preserves the distances of vectors in $\mathcal{V}$ with distortion factor $(1 \pm \delta_{JL})$. This guarantees that if a good solution exists in $\mathcal{V}$, a projection of it exists in $\mathcal{S}$ with bounded error.

Thus, optimizing $\eta$ in the low-dimensional space $\mathbb{R}^a$ is theoretically guaranteed to find a solution close to the optimal manifold $\mathcal{V}$, provided $a > d_{int}$. $\qquad \square$

**Discussion.** This theorem explains why RFID-MoE achieves superior performance with a very small parameter budget (e.g., 3% of original parameters). The residual $R$ does not need to be reconstructed exactly (which would require $D$ parameters); instead, we only need to reconstruct its projection onto the intrinsic solution manifold $\mathcal{V}$. Our parameter-efficient $\eta$ is sufficient to cover this intrinsic dimension.

