# OpenReview forum: "Effective MoE-based LLM Compression by Exploiting Heterogeneous Inter-Group Experts Routing Frequency and Information Density"
_ICML.cc/2026/Conference — ICML 2026 regular_

### Official Review · Reviewer_bQaa · 2026-02-28

**Soundness:** 3
**Presentation:** 3
**Significance:** 2
**Originality:** 3
**Overall Recommendation:** 4
**Confidence:** 3

**Summary:**

This paper introduces RFID-MoE, a post-training compression framework for Mixture-of-Experts (MoE) LLMs. It performs adaptive rank allocation by combining routing frequency and effective rank, paired with parameter-efficient residual reconstruction. Experts are grouped by activation frequency, and their importance scores integrate normalized routing frequency and entropy-based effective rank to assign non-uniform rank budgets for SVD-based basis sharing. To reduce residual error, it uses a sparse orthonormal random projection from a low-dimensional trainable vector to the expert weight space, shared across groups. Experiments on various MoE backbones show consistent gains over recent decomposition baselines (e.g., MoBE), especially under high compression ratios.

**Compliance With Llm Reviewing Policy:**

Affirmed.

**Final Justification:**

The authors have addressed my concerns. However, I believe that some issues in the paper still remain unresolved. Therefore, I will keep the score unchanged.

**Key Questions For Authors:**

1. How are compression ratios calculated when residual vectors are included? Do the reported ratios account for the full storage of A, B, α, η, and any indices for P? How are the ranks adjusted to ensure the total storage stays within the predefined budget?

2. What are the actual memory footprints (in GB) and inference throughput (tokens/sec, TFLOPs) before and after compression for representative models? How does the residual reconstruction influence runtime efficiency and memory locality?

3. How sensitive is the model performance to group size \(k\), number of groups \(m\), and residual dimension \(a\)? Could you provide ablation studies by varying these hyperparameters, and report the marginal accuracy gains per additional percentage of residual parameters?

**Limitations:**

The author did not fully discuss the limitations and potential negative social impacts of their work. Please add them in the revised version.

**Strengths And Weaknesses:**

### Strengths
1. Combining dynamic routing frequency with an information-density metric (effective rank) for adaptive rank allocation is well-founded and effectively overcomes the limitations of uniform or purely static allocation strategies.
2. Comprehensive evaluations are conducted across various MoE architectures (Qwen3-30B, Qwen2-57B-A14B, DeepSeekMoE-16B-Base, Qwen1.5-MoE A2.7B, along with lightweight models), using both perplexity and zero-shot reasoning as evaluation metrics.
3. The motivation—addressing heterogeneous routing patterns and non-negligible residual errors—is convincing and reinforced by visualizations of routing distribution and residual spectra.

### Weaknesses
1. The theoretical analysis mainly restates isometric properties (P^T P = I) and high-level intrinsic dimensionality claims. The proposed ε-approximation theorem is informal and lacks concrete, verifiable guarantees for the random one-hot projection.
2. Runtime and memory efficiency results are missing, including compressed model size (with residuals), actual compression time, inference throughput, and FLOP variations. Such metrics are essential for compression studies, ideally presented in quality–efficiency trade-off plots.
3. A more thorough comparison with residual-aware SVD methods (e.g., ResSVD-based two-stage truncation) and UniLoRA-style single-vector residual schemes would better position the proposed residual mechanism.

---

> ### Author Rebuttal · Authors · 2026-03-30
>
> **W1. Theoretical analysis of Residual reconstruction**
>
> ​​Thanks for reviewer's suggestion. Our analysis provides an intuitive justification for residual reconstruction rather than a strict theoretical guarantee. The approximate isometry property $P^\top P = I$ preserves residual geometry in expectation, consistent with prior work on random projections (e.g., UniLoRA). Empirically, Table 2 shows that adding residual reconstruction for SVD compression consistently improves performance under high compression ratio, validating its practical effectiveness.
>
> From an experimental perspective,  We further conduct experiments to decouple adaptive rank and residual components, and the results are summarized in table below. We observe that introducing the residual vector significantly improves performance.
>
> **Effect of Residual Vector under 50% Compression (Qwen3-30B-A3B-2507)**
>
> | Setting| WikiText PPL ↓ | PTB PPL ↓ |
> |-------|-----|-----|
> | MoBE baseline | 10.00 | 17.41|
> | With Residual Vector (Ours)| 8.02 | 13.53 |
> | Without Residual Vector (Ours)  | 8.91 | 15.06 |
>
> **W3. Comparison of residual reconstruction with ResSVD and UniLoRA**
>
> ​​Thanks for reviewer's suggestion. We would like to clarify that our method differs from ResSVD and UniLoRA in several key aspects.
> First, unlike ResSVD which targets dense LLMs, our method is specifically designed for MoE models and accounts for heterogeneous expert utilization by incorporating both routing frequency and information density.
> Second, instead of performing a second-stage SVD for residual approximation, we model the residual using a low-dimensional latent vector with sparse projection, which avoids additional decomposition overhead and is more parameter-efficient.
> Third, rather than using a fixed rank split, we adopt adaptive rank allocation across experts, enabling better preservation of task-relevant information.
> Finally, UniLoRA focuses on parameter efficiency in LoRA-based fine-tuning, whereas our method targets residual reconstruction in SVD-based compression. These approaches address fundamentally different problems and are therefore not directly comparable.
>
> **Q1. Compression ratio calculation**
>
> The compression ratio is computed as the total number of parameters in A, B, and η divided by the total number of parameters across all experts.
> We clarify that the reported compression ratio includes all parameters in A, B, and the residual vector η. Given a target compression ratio, we first reserve a fixed budget for η, and then perform adaptive rank allocation within the remaining budget.
>
> The projection matrix P is shared across layers and introduces negligible overhead (e.g., 0.06% in Qwen3-30B), while the blending coefficients α account for only 0.001%.
>
> Rank allocation is performed at the group level by computing coefficients Cg (Eq. 12) and assigning ranks Kg proportionally (Eq. 13).
>
>
>
> **W2 & Q2: Memory footprints  and inference throughput**
>
> ​​Thanks for reviewer's suggestion. We observe that under a 60% compression ratio, our method reduces the memory footprint of the Qwen3-30B model to approximately 37 GB, compared to about 57 GB for the original model.
>
> We follow matrix factorization-based compression baselines (such as MoBE and D2-MoE), which simulate factorization via multiple calls to existing fused MoE kernels, to measure inference speed.
> Our method can reduce latency to achieve comparable throughput to the MoBE baseline when activating fewer experts, while delivering better perplexity. For example, on Qwen3-30B at a 50% compression ratio, MoBE achieves 743 tokens/s with a PTB perplexity of 17.41, whereas our method, with only 4 active experts, attains a lower perplexity of 15.21 with a similar throughput of 712 tokens/s.
>
>
>
> **Q3: Ablation for k, m, and dimension a**
>
> ​​Thanks for reviewer's suggestion. We further provide an ablation study on group size and the number of groups using the Qwen3-30B model in Table below (40% compression). The product of group size k and the number of groups m is fixed to the number of experts per layer (128).
>
> | Group Size (k) | Groups (m) | PTB PPL ↓ | WikiText PPL ↓ |
> |-----|----|----|----|
> | 2  | 64 | 14.04 | 8.55|
> | 4  | 32 | 12.81 | 7.59|
> | 8  | 16  | 12.85| 7.59|
>
> These results indicate that performance becomes stable when the group size exceeds 4, while grouping 4 experts per group achieves slightly better performance.
>
> We also include an ablation study on the residual vector dimension relative to the dense model dimension, as shown in the table below (Qwen3-30B 50% compression). The results indicate that using a residual dimension of 3% achieves the best performance.
>
> | Residual Dim (%) | WikiText PPL ↓ | PTB PPL ↓ |
> |------|----|---|
> | MoBE baseline    | 10.00  | 17.41 |
> | 0% (no residual) | 8.91  | 15.06 |
> | 1% | 8.52  | 14.54  |
> | 3% | 8.02  | 13.53 |
> | 5% | 9.90 | 16.52 |

---

> > ### Author Rebuttal · Reviewer_bQaa · 2026-04-03
> >
> > The authors have addressed my concerns. However, based on the overall innovation and feedback of the paper, I tend to maintain the original score.

---

### Official Review · Reviewer_cgGk · 2026-03-12

**Soundness:** 2
**Presentation:** 2
**Significance:** 2
**Originality:** 3
**Overall Recommendation:** 4
**Confidence:** 4

**Summary:**

The paper proposes a new method for allocating ranks in SVD‑based MoE compression, together with a error‑compensation mechanism. The rank allocation strategy is formulated as a convex combination of routing frequency and the effective rank of each expert and its weight. After applying SVD with the allocated ranks, the method introduces an additional error‑compensation term constructed using a shared projection matrix and individually learned correction vectors to further reduce approximation error. Experiments conducted across different architectures and baselines demonstrate that the proposed method consistently outperforms previous methods.

**Compliance With Llm Reviewing Policy:**

Affirmed.

**Final Justification:**

The authors have addressed my main concerns in the rebuttal, and I will adjust my score accordingly.

**Key Questions For Authors:**

1. As highlighted in the weakness section, could the authors address the concern about insufficient empirical support?
2. I could not find any discussion of limitations. The paper should include a dedicated limitations section, or at least explicitly mention key limitations.

**Limitations:**

A dedicated limitations section is missing. Adding one that clearly articulates the method’s assumptions and potential failure cases would make the paper more complete and help practitioners better understand when the approach is most appropriate.

**Strengths And Weaknesses:**

Strengths:
1. Emphasizing effective rank helps preserve the capacity of rarely activated yet critical experts, the resulting heterogeneous rank allocation improves quality over uniform compression.
2. The method outperforms prior approaches overall across reported settings.
3. The work introduces an error compensation term, which is an aspect often overlooked in prior SVD‑based compression methods.

Weaknesses:
1. While a theoretical motivation is provided, there is no clear ablation isolating the error compensation component. A direct comparison with vs. without the compensation term would strengthen the empirical case.
2. Because the approach relies on gradient-based optimization and reconstructs weights from compressed metadata, the end-to-end compression cost and the runtime overhead at inference should be reported and compared against baselines.
3. The paper notes that some experts receive zero activations and proposes groupwise design to mitigate this, but it does not quantify whether the smallest groups still suffer zero activation or how frequently this occurs. For example, Table 1 does not report activation frequencies for the rarest groups (e.g., groups 31 and 32), making it difficult to assess potential over compression in the long tail.
4. Although the method aims to preserve rarely used experts, the experiments focus on common‑sense QA and perplexity, where such experts are seldom triggered. Additional evaluations on specialized domains, such as mathematics, would better substantiate the claim that critical, rare experts are preserved.

---

> ### Author Rebuttal · Authors · 2026-03-30
>
> We thank the reviewer for the detailed feedback. We address each concern below.
>
> **W1. Ablation isolating the error compensation component**
>
> **Effect of Residual Vector under 50% Compression (Qwen3-30B-A3B-2507)**
>
> | Setting                  | WikiText PPL ↓ | PTB PPL ↓ |
> |--------------------------|---------------|-----------|
> | MoBE baseline | 10.00         | 17.41     |
> | adaptive rank allocation(Ours)  | 8.91          | 15.06     |
> | adaptive rank allocation + Residual Vector (Ours)     | 8.02          | 13.53     |
>
> We further conduct experiments to decouple adaptive rank and residual components, and the results are summarized in the table above. Specifically, on PTB, our rank allocation can improve PPL from **17.41** to **15.06**, and further incorporating residual vector can lead to **13.53** PPL.
>
>
> **W2. compression cost and inference overhead**
>
> For the Qwen3-30B model, compressing and reconstructing the gate and up matrices for a single layer takes approximately 34 minutes using our method, compared to around 29 minutes for the MoBE baseline. The peak GPU memory usage during compression is about 8.3 GB.
> Importantly, our compression procedure is performed as a one-time offline process, and does not introduce any additional overhead during online inference. In addition, under a 60% compression ratio, our method reduces the memory footprint of the Qwen3-30B model to approximately 37 GB, compared to 57 GB for the original model.
>
> We follow matrix factorization-based compression baselines (such as MoBE and D2-MoE), which simulate factorization via multiple calls to existing fused MoE kernels, to measure inference speed.
> Our method can reduce latency to achieve comparable throughput to the MoBE baseline when activating fewer experts, while delivering better perplexity. For example, on Qwen3-30B at a 50% compression ratio, MoBE achieves 743 tokens/s with a PTB perplexity of 17.41, whereas our method, with only 4 active experts, attains a lower perplexity of 15.21 with a similar throughput of 712 tokens/s.
>
> **W3. Groupwise design to mitigate zero activation**
>
> Many thanks for the question. We would like to clarify that **our paper does not claim that the grouping strategy is designed to mitigate zero-activation experts**. Instead, our grouping design is inspired by prior works such as Basis Sharing and MoBE, where grouping enables multiple weight matrices to share a common basis, leading to more expressive representations and improved performance.
> For the concern about low-activation groups, grouping may help address the zero activation issue since the least-activated group still receives non-zero activations in practice. For example, in the first layer of the Qwen3-30B model, Group 32 (the least-activated group) has a total activation frequency of 55. This may lead to the misunderstanding. However, we highlight that grouping is designed to incorporate multiple weight matrices to share a common basis for better compression performance.
>
> Note that our key point in the paper is that low activation frequency does not necessarily imply low importance. We explore the rank allocation for experts of various frequency, rather than balancing the activation frequency of experts.
> Specifically, in our paper, we observe that expert activation in MoE models is highly imbalanced, with some experts receiving extremely low or even zero activations. **However, low activation frequency does not necessarily imply low importance, as such experts may still contain substantial intrinsic information (i.e., high information density)**.
> To address this mismatch, we jointly consider activation frequency and intrinsic information content (measured by effective rank) to assess expert importance. This allows us to allocate non-zero rank even to infrequently activated experts, avoiding their collapse into zero-rank representations.
> For example, although Group 32 has a total activation frequency of only 55, our method still assigns a rank of 460 under a 40% compression ratio, demonstrating that low-frequency experts can still retain significant capacity when guided by effective rank.
>
> **W4 & Q1. More experiments such as math**
>
> Following the reviewer’s suggestion, we include additional experiments on three mathematical reasoning datasets and a large-scale multi-domain benchmark (MMLU).
> Under a 60% compression ratio on the Qwen3-30B model, Table below shows the comparison of our method with MoBE baseline.
>
> | Dataset  | MoBE Acc ↑ | Ours Acc ↑ |
> |----------|------------|------------|
> | MathQA   | 0.51       | 0.52       |
> | GSM8K    | 0.53       | 0.62       |
> | arithmetic    | 0.32       | 0.35       |
> | MMLU     | 0.67       | 0.69       |
>
> **Q2: Limitations**
>
> We thank the reviewer for the suggestion. We will include a discussion of limitations in the revision, including the potential challenges for application to VLM-based MoE models.

---

> > ### Author Rebuttal · Reviewer_cgGk · 2026-04-03
> >
> > The authors have addressed my main concerns in the rebuttal, and I will adjust my score accordingly.

---

### Official Review · Reviewer_Z7Fn · 2026-03-13

**Soundness:** 2
**Presentation:** 3
**Significance:** 3
**Originality:** 2
**Overall Recommendation:** 4
**Confidence:** 3

**Summary:**

This paper proposes RFID-MoE, a compression framework for Mixture-of-Experts (MoE) large language models that addresses two challenges in SVD-based expert compression. First, the authors introduce an adaptive rank allocation strategy that combines expert routing frequency with "effective rank" (a spectral entropy-based measure of information density) to assign higher compression ranks to more important expert groups. Second, they propose a parameter-efficient residual reconstruction mechanism using a sparse isometric projection matrix to recover information lost during low-rank approximation. Experiments are conducted on several MoE models (Qwen3-30B, DeepSeekMoE-16B, Qwen2-57B, etc.) at compression ratios of 20%–60%, showing improvements over baselines such as MoBE and D2-MoE on perplexity and zero-shot reasoning benchmarks.

**Compliance With Llm Reviewing Policy:**

Affirmed.

**Final Justification:**

I thank the authors for the detailed rebuttals backed with new empirical results.

The empirical evidence and the failure mode analysis have addressed my concern regarding additive vs. multiplicative fusion.

The direct comparison with REAP on Qwen3-30B is convincing.

So I will increase my score accordingly.

**Key Questions For Authors:**

1. Is residual reconstruction optimized jointly with the low-rank factors, or sequentially? What is the exact training loss?

**Limitations:**

No, the paper has no dedicated limitations section.

**Strengths And Weaknesses:**

## Strengths

1.  The observation that expert routing frequencies are highly imbalanced and that compression residuals contain non-negligible energy is well-documented and provides clear motivation.

2. The main empirical results are reasonably broad in model coverage and show consistent improvements over several recent baselines in perplexity, with some notably large gains on Qwen3 at high compression ratios.

## Weaknesses

1. Each individual component (effective rank metric, sparse isometric projection, routing frequency metric) draws heavily from prior work. The paper would benefit from a deeper discussion of why this particular combination is principled rather than one of many possible heuristic fusions.

2. The paper does not cleanly separate the gains from (a) adaptive rank allocation alone versus (b) residual reconstruction alone. Table 4 ablates ξ but always includes residual reconstruction.

3.  The paper does not report compression time, training time for residual reconstruction, or memory overhead during the compression pipeline itself.

4.  The paper omits important concurrent methods, most notably REAP (Lasby et al., 2025, https://arxiv.org/abs/2510.13999), a strong pruning method that achieves near-lossless compression at 50% on overlapping models like Qwen3-30B. MoNE (Zhang et al., 2025, https://arxiv.org/abs/2507.00390), which replaces pruned experts with lightweight surrogates using frequency and output variance metrics, is also missing and directly challenges RFID-MoE's assumption that all experts should be retained at reduced rank.

---

> ### Author Rebuttal · Authors · 2026-03-30
>
> **W1. Components of our work**
>
> Thanks for reviewer's suggestion. First, we observe that expert activation in MoE models is highly imbalanced. However, low activation frequency does not necessarily imply low importance. In practice, we find a mismatch between activation frequency and the intrinsic information content of experts. Therefore, we introduce the effective rank to quantify the information density of each expert, and combine it with activation frequency to perform adaptive rank allocation, leading to more effective compression performance.
> Second, we observe that most existing compression methods directly discard the residual component after low-rank approximation. However, the residual still contains non-negligible information and can be efficiently reconstructed with a small number of parameters. To this end, we adopt an isometric projection to capture and reconstruct this informative residual component.
>
> Therefore, our method does not simply combine existing techniques, but rather a framework motivated by specific observations on MoE models.
>
> **W2. Ablation for Residual**
>
> **Effect of Residual Vector under 50% Compression (Qwen3-30B-A3B-2507)**
>
> | Setting  | WikiText PPL ↓ | PTB PPL ↓ |
> |----|----|----|
> | MoBE baseline | 10.00 | 17.41 |
> | adaptive rank allocation (Ours)  | 8.91 | 15.06 |
> | adaptive rank allocation + Residual Vector (Ours) | 8.02 | 13.53 |
>
> We further conduct experiments to decouple adaptive rank and residual components, and the results are summarized in the table above. Specifically, on PTB, our rank allocation can improve PPL from **17.41** to **15.06**, and further incorporating residual vector can lead to **13.53** PPL.
>
> **W3. Compression, training time and memory**
>
> For the Qwen3-30B model, compressing and reconstructing the gate and up matrices for a single layer takes approximately 34 minutes using our method, compared to around 29 minutes for the MoBE baseline. However, our compression procedure is operated offline with just a one-time cost, and does not introduce any additional overhead during online inference. The peak GPU memory usage of both methods during compression is about 8.3 GB. We believe this compression cost is acceptable that enables compressing a 30B model on most GPUs with memory above 10GB.
>
> **W4. REAP and MoNE**
>
> We thank the reviewer for pointing out these relevant works and we will cite and discuss these papers in our revision. MoNE prunes low-importance experts based on activation frequency and output variance, and replaces them with lightweight surrogate modules to reduce memory cost while maintaining performance. REAP performs router-aware expert pruning by leveraging gate values and activation statistics, achieving near-lossless compression by selectively removing less important experts.
> Unlike MoNE and REAP, which remove low-importance experts via pruning, **we perform continuous, adaptive compression by jointly considering activation frequency and intrinsic information content (measured by effective rank). This allows us to avoid collapsing low-frequency experts to zero rank while still achieving effective compression.**
>
> **Assumption:** In our paper, we highlight a key limitation of relying solely on routing frequency for compression. ‘’While incorporating routing frequency appears intuitive, activation distributions in large MoE models are often highly skewed, with some experts receiving near-zero activations. As a result, compression strategies that allocate ranks purely based on frequency may assign zero rank to certain experts, effectively degenerating into expert pruning.’’ This can lead to performance degradation, especially on specialized tasks where infrequently activated experts may encode important domain-specific knowledge.
> We would like to clarify that our claim specifically targets methods that rely solely on routing frequency. In contrast, recent works such as MoNE and REAP introduce additional signals (e.g., output variance in MoNE, and gate values and activation statistics in REAP) to improve pruning decisions beyond frequency alone, which do not challenge our claim.
> Moreover, prior work such as D²-MoE explicitly points out that “expert pruning methods can significantly reduce model size, but often suffer from irreversible loss of expert knowledge, which degrades model performance and typically requires additional fine-tuning to recover.”
>
> **Q1. Residual optimized jointly and loss**
>
> Yes, the residual component and the retained low-rank component are optimized jointly. During the reconstruction process, P is fixed, while both the vector ξ and the low-rank components are treated as trainable parameters. We present the scaled loss curves in **Figure 4**. Specifically, the results are obtained on the 47-th layer of the Qwen3-30B-A3B-2507 model. For example, for compressing the up projection matrix, the training loss of MoBE drops from about 0.16 to 0.098, whereas RFID-MoE achieves a lower convergence training loss at 0.08.

---

> > ### Author Rebuttal · Reviewer_Z7Fn · 2026-04-02
> >
> > - 1. Why is a simple linear interpolation (Eq. 12, $C_g = \xi * E_g + (1 - \xi) * F_g$) the right way to fuse routing frequency and effective rank? What about multiplicative fusion, or a learned weighting? Is there any theoretical or empirical evidence that this specific design is superior to alternative fusion strategies?
> >
> > - 2. Can you provide a direct empirical comparison with REAP on Qwen3-30B at 50% and 60% compression ratios? Since REAP reports results on this exact model, a side-by-side comparison would be the most convincing way to address W4. If pruning-based methods with complementary signals (output variance in MoNE, gate values in REAP) achieve comparable or better results, then the paper's core narrative that all experts must be preserved at reduced rank may be weakened.

---

> > > ### Author Response · Authors · 2026-04-03
> > >
> > > **[About the additive formulation]**
> > >
> > > Thanks for reviewer's question.
> > > Empirically, additive formulation is more suitable for our rank allocation setting. As pointed out in prior work [1] which distinguishes additive and multiplicative formulations, additive formulations measure whether the joint effect exceeds the sum of individual contributions, and are often more appropriate for decision-making and more suitable for allocation settings (e.g. ranks for expert groups). In contrast, multiplicative formulations capture interaction effects between factors and reflect dependencies among them. We believe that both the routing frequency and the normalized effective rank are key factors for importance evaluation, thus we adopt the additive formulation.
> > >
> > > In practice, we experiment with the multiplication formulation where we replace Equation (12) with $\xi * E_g * (1-\xi)*F_g$.
> > > The following table shows the performance comparison under addition or multiplication formulation at a 50% compression ratio on the Qwen3-30B model. As shown in the table, using multiplication for rank allocation is less effective than using addition.
> > >
> > > | Setting                  | WikiText PPL ↓ | PTB PPL ↓ |
> > > |--------------------------|---------------|-----------|
> > > | MoBE baseline | 10.00         | 17.41     |
> > > | Multiplicative  |     9.52      |   16.06   |
> > > | Additive (Ours)     | 8.02          | 13.53     |
> > >
> > >
> > > Furthermore, we observe that with multiplicative formulation in Equation (12),  Group 30 in Table 1 is allocated only **20** ranks, despite having an effective rank as high as **1049**. In contrast, the additive approach manages to allocate **475** ranks to Group 30. Therefore, additive fusion better ensures that experts with higher information density are allocated a relatively larger number of ranks, thereby maintaining a better balance between frequency and effective rank.
> > >
> > > Thanks for the suggestion with learned weighting. Using learned weighting will introduce additional trainable parameters with extra overhead, thereby increasing the training cost. Our current fixed weighting addition already demonstrates the effectiveness of combining  the routing frequency and the normalized effective rank. We leave the exploration with learned weighting as our future work.
> > >
> > >
> > >
> > > **[Comparison with REAP]**
> > >
> > > **Comparison of accuracy under 25% Compression (Qwen3-30B)**
> > >
> > > | Dataset  |  MMLU  |  WinoG. | Hellaswag |  ARC_c |
> > > |----------|------------|------------|------------|------------|
> > > | REAP   | 0.67       | 0.70       | 0.70       | 0.48       |
> > > | MoBE    | 0.78       | 0.73       | 0.77      | 0.63       |
> > > | RFID-MoE (Ours)   | 0.80      | 0.73       | 0.78       | 0.64       |
> > >
> > > **Comparison of accuracy under 50% Compression (Qwen3-30B)**
> > >
> > >
> > > | Dataset  |  MMLU  |  WinoG. | Hellaswag |  ARC_c |
> > > |----------|------------|------------|------------|------------|
> > > | REAP   | 0.50       | 0.58       | 0.48       | 0.35       |
> > > | MoBE    | 0.73       | 0.72       | 0.68      | 0.59       |
> > > | RFID-MoE (Ours)   | 0.75       | 0.73       | 0.74       | 0.60       |
> > >
> > > The tables above present the accuracy of REAP, MoBE baselines, and our method across four datasets using the Qwen3-30B model at 25% and 50% compression ratios (REAP results are sourced from Table A6 in the original publication). As observed, the accuracy of REAP is lower than that of both MoBE and RFID-MoE. In contrast, our method outperforms the MoBE baseline, achieving an accuracy improvement of up to 6% on the  Hellaswag dataset at 50% compression ratio. This comparison reinforces our discussion regarding Challenge 1 in Section 1: relying solely on frequency to evaluate expert importance and subsequently removing those experts that are rarely activated may result in severe performance degradation on specialized tasks where infrequently-activated experts encode critical domain-
> > > specific knowledge.

---

### Decision · Program_Chairs · 2026-04-30

**Decision:**

Accept (regular)

**Comment:**

Based on the three reviews, all of which rate the paper as “Weak Accept,” the consensus is that this is a technically solid contribution to MoE compression with clear empirical improvements. The reviewers appreciate the adaptive rank allocation combining routing frequency and effective rank, as well as the residual reconstruction mechanism. However, consistent concerns emerge regarding the lack of ablations isolating the contribution of each component, missing runtime and memory efficiency metrics, insufficient comparison with concurrent methods (e.g., REAP, MoNE), and the absence of a limitations section. While the paper advances an important sub-area and provides broad experimental validation, these weaknesses temper its current impact. Therefore, I recommend a Weak Accept.